# Deaf children with cochlear implants in Chile: A national analysis of health determinants and outcomes in the Latin American context

**Mario Bustos-Rubilar** [1,2]*, **Fiona Kyle**[1], **Merle Mahon**[1]

1 Division of Psychology and Language Science, University College London, London, United Kingdom,
2 Departamento de Fonoaudiología, Facultad de Medicina, Universidad de Chile, Santiago, Chile

* mario.rubilar.18@ucl.ac.uk

## Abstract

Deafness from birth represents a critical challenge for children's communication, with substantial public health considerations. One intervention has been cochlear implants (CI) for children with severe to profound deafness. Since 2008, Chile has implemented regulations to provide a CI at an early age. However, wide variability exists in factors and expected outcomes without previous national studies. This study aimed to characterise deaf children with CI in Chile and evaluate the impact of CI on speech perception and production, social inclusion, and parental satisfaction.

We conducted a prospective study using hospital clinical records and an online questionnaire with 107 deaf children under 15 who had received CIs from 2017 to 2019. We characterised factors and outcomes and investigated the relationship between demographic, audiological, and social determinants of health and outcomes, including communication at home, CAPII, SIR, Geers and Moog Scale, Social Inclusion, and Parental Satisfaction.

Our study showed a national profile of deaf children with CI, representing 70% of those implanted from 2017 to 2019. CI beneficiaries lived in more developed boroughs (.54) compared to the national average (.37). Communication and speech perception outcomes varied and were concerning, yet more positive outcomes were presented for social inclusion and parental satisfaction. We found an association between the measured outcomes and children's age, a socio-economic factor, CI use and CI training. This novel national study supports integrating public services close to each beneficiary's borough to improve outcomes with the device. CI use and parental training might be crucial measures during rehabilitation treatment.

## 1. Introduction

Deafness affects population health worldwide, having significant economic costs and health consequences [2]. The global incidence of severe to profound deafness from birth is over

**Data availability statement:** Data is available to other researchers on The Center for Open Science (COS) at the following link: https://osf.io/4hfd9/

**Funding:** Data collection was partially supported by a small grant from the Chilean Ministry of Health through National Public Tender 757-89-I120. This article was also supported by a full scholarship provided by the Chilean Government, "Beca de Doctorado en el Extranjero Becas Chile, Convocatoria 2018, Ley N°21.053, Asociación Nacional de Investigación y Desarrollo (ANID)" and the Chilean Ministry of Health through National Public Tender 757-89-I120. The funders had no role in study design, data collection and analysis, decision to publish, or preparation of the manuscript.

**Competing interests:** The authors declare no conflict of interests.

430 million people globally. [3]. In Latin America, about 16 million children under 15 are deaf [4,5]. We use the inclusive term "deaf children" to cover the spectrum of hearing loss in children, including those who are hard of hearing. In Chile deafness occurs in 2.8 per 1000 newborns, with school-aged children's prevalence ranging from 0.2% to 7.8% [6]. The cost of unaddressed deafness in childhood is estimated at USD 750-790 billion in the region, according to the World Health Organization [4]. Importantly, there is wide variability in medical interventions for deaf children within Latin American countries, and no extensive cost analyses have been performed for this region.

Deaf children with severe to profound deafness from birth or early age face significant challenges in language development, impacting their social inclusion and communication [7]. Unaddressed deafness can cause significant damage in language development, education progress, social inclusion, economic dependency and future possibilities of skilled jobs [7,8]. Without exposure to accessible sign language or appropriate rehabilitation strategies, deaf children's quality of life, social inclusion, and communication skills are highly compromised, especially in low-resource areas, such as in most Latin American settings [2]. Moreover, detrimental conditions for deaf children intervention are found in Latin America compared to high-income countries. Lack of universal access to healthcare, under-resourced hospital infrastructure, area-level deprivation, living far from hospital services, high technology costs, and unequal distribution of healthcare professionals impact well-being and contribute to socioeconomic inequalities [8].

For deaf children, early intervention is crucial and is informed by two models of disability: the medical model, focusing on auditory and spoken language development [9], and the socio-anthropological model, which emphasizes recognizing and promoting deaf culture and sign language [10]. In high-income countries, early intervention follows the 1-3-6 guidelines: screening by one month, diagnosis by three months, and intervention by six months [11]. Although a multidimensional approach including sign language acquisition, and social and educational consideration is widely recommended [12], early provision of hearing devices such as CIs is usually the central and dominating recommendation in the medical mode. Using CIs is an effective, although costly, intervention for deafness and consequent improvements in deaf children's health status, spoken language development and social inclusion [11]. Nevertheless, there is a lack of evidence of the effectiveness of CIs in middle- and low-income regions compared to developed countries, especially considering CI is a high-cost treatment in health policies.

Policy recommendations for deaf children vary by country. High-income countries often provide comprehensive CI services under universal health coverage [13]. Middle and low-income countries, where most deaf children live, face challenges in policy coverage and cost efficiency. In Latin America, countries like Mexico, Argentina, and Brazil have national CI policies, but recent monitoring is lacking [1,14,15]. In Chile, one of the most developed countries in the region in terms of health, education and access to services, deafness was included in the 2013 "Explicit Healthcare Guarantees - GES" regulation, providing standardised diagnosis and treatment criteria for children. As part of this rehabilitation programme, families apply for the intervention under a multistep selection process based on international recommendations. However, access to appropriate rehabilitation can be limited, for example for some children within high deprivation social contexts due to the lack of potential support during the rehabilitation process.

Similarly, very few hospitals are included as CI centres in the country, thus forcing those children not living in main urban areas to receive the rehabilitation service far away from their homes. The CI provision has been also expanded to include children and adults but with very recent national screening only available from 2020 [16].

Among children with CI, the evaluation outcomes often consider speech perception tests, language development, parental satisfaction and social inclusion evaluations, which in many cases are completed by parents and caregivers [17]. Nevertheless, children's results might vary depending on social determinants of health, their family's Borough Development Index (BDI) and other factors, such as level of education about the health condition. Identifying factors associated with successful outcomes is critical. These factors include early diagnosis and rehabilitation, early use of hearing aids (HA), absence of comorbidities, properly functioning CI, consistent device use, family education and engagement, and access to rehabilitation services [18].

Characterisation studies in other countries, including the UK, US, India, China, Malaysia, Turkey, and Japan have provided valuable data on demographic, socioeconomic, epidemiological, and audiological factors and outcomes regarding children with CI [19,20]. However, the Latin American region lacks current national characterisations, reducing the possibility of monitoring factors and outcomes expected for deaf children with CIs. This study will characterise deaf children with CI in Chile and evaluate the impact of the CI in speech perception and production, social inclusion, and parental satisfaction.

## 2.  Material and methods

### Data source

We conducted a prospective study consisting of an audit of hospital clinical records and a parental survey, which was conducted between September 2020 to March 2021. Thus, we gathered data from two main sources: 1) Hospital clinical records from adults attending public hospitals in Chile (refer to "Appendix 2"). 2) An online survey ("Appendix 3") completed by each of the parent/closest caregiver of each child. A written informed consent was completed for each respondent at the beginning of the online survey.

A bilingual committee of English and Spanish speakers developed the survey, adapted it for online use and distributed it via email or text to the closest caregiver of each participant. To ensure accessibility, we provided various support measures, including sign language interpreters, video calls, and telephone facilitators as needed. The survey included two formal parent report assessments – Chilean Version of the Categories of Auditory Performance CAP II (CAPII) and Chilean Version of the Speech Intelligibility Rating Scale SIR (SIR)– which were included in the survey and are available in Table 1.

### Population

We invited all parents and caregivers of children under 15 years of age who had received CI between 2017 and January 2019 in all the available centres in the country. One hundred and seven children were included in the study (see consort diagram "Appendix 1"), representing 70% (107 out of 153) of all children implanted in the public health system in Chile during the mentioned period. It is important to consider that, although universal newborn screening for hearing loss has been mandatory since 2020, participants in the current study were not covered by this, resulting potentially in delays in their diagnosis of deafness. Exclusion criteria: children who had been implanted less than one year at time of recruitment and those from private institutions (approximately 18% of deaf children with CI). The reason for the latter exclusion criterion was that the majority of deaf children with CIs with vulnerable backgrounds and potentially more social determinants affecting their outcomes were treated under the public system. Therefore, evaluation of the public system has a significantly greater impact in terms of equity and reach. Table 1 summarises the variables included in our analyses with their respective data source descriptors.

**Table 1. Dependant and independent variables.**

| Variable | Description | Source |
|---|---|---|
| Dependent Variables | | |
| Communication at home [21] | Four options for the question: How do you communicate with your child?: 1) Sign Language, 2) Spoken Language, 3) Mixed, using Sign and Spoken Language, 4) Other. | DHH Survey |
| Geer and Moog Latin American Categories of Speech Perception. [22] Abb = GeersM | Eight options: 0) Unable to perceive any speech sound (Ling test), 1) Speech detection but unable to perceive even time-intensity pattern info in words with speech amplified, 2) Time-intensity pattern perception in amplified speech (Chilean ESP Test: Above 70% in testing with Chilean ESP), 3) Word identification by basic spectral information (Chilean PIP test: words with same metric and time, but different consonants and vowels), 4) Word identification in a group by their vowels (Chilean PIP Test, 5) Recognition of words in a group by their consonants (Chilean PIP test), 6) Word recognition by repetition (Chilean word repetition test), 7) Spoken Language Comprehension (Chilean information evaluation of spoken language comprehension). | Clinical Record |
| Categories of Auditory Performance CAP II (Chilean version of CAPII) [23] Abb = CAPII | Ten levels in the scale: 0. No awareness of environmental sounds 1. Awareness of environmental sounds 2. Response to speech sounds 3. Recognition of environmental sounds 4. Discrimination of at least two speech sounds 5. Understanding of common phrases without lip-reading 6. Understanding of conversation without lip-reading with a familiar talker 7. Use of a telephone with a familiar talker8. Understanding/Following group conversations. 8. Use the telephone with an unknown speaker in an unpredictable context. | DHH Survey |
| Speech Intelligibility Rating Scale SIR (Chilean Version of SIR) [23] Abb = SIR | Six levels in the scale: 1. Connected speech is unintelligible. Pre-recognizable words in spoken language, the child's primary mode of everyday communication may be manual. 2. Connected speech is unintelligible; intelligible speech is developing in single words when context and lip-reading cues are available 3. Connected speech is intelligible to a listener who concentrates and lip-reads within a known context. 4. Connected speech is intelligible to a listener who has little experience of a deaf person's speech. 5. Connected speech is intelligible to all listeners. The child is understood easily in everyday contexts. | DHH Survey |
| CI Satisfaction, | Five level Likert item: *Has the use of the device satisfied your expectations as a parent/caregiver?*: 1 (Unsatisfied), 2, 3, 4, 5 (Very Satisfied). | DHH Survey |
| Social Inclusion | Five level Likert item: *Do you feel that your child is included in school and social life?* 1 No, not too much included, 2,3,4, 5 Yes, the child is quite included. | DHH Survey |
| Independent Variables | | |
| Age | In months | Clinical Records |
| Gender | 1) Female, 2) Male | Clinical Records |
| CI age | Age in months at CI surgery. Chronological age when the Children with CI have the CI surgery/Switch-On. In this study age at CI surgery and at Switch-On are considered the same due to the Chilean regulation, which requires the CI Switch on within one month after the CI surgery. | Clinical Records |
| Deafness presentation | 1) Congenital, 2) Late Deafness ^ | Clinical Records |
| SHI | Socioeconomic health insurance: 1) Low income, 2) Low- middle income, 3) Middle income, 4) Middle, high income. This public health insurance covers 82-83% of the entire Chilean population. People might incur proportional out-of-pocket expenditure for CI provision according to this classification. | Clinical Records |
| BDI | Borough Development Index: Index from 0.0 (low) up to 1.0 (high) related to each territory's socioeconomic outcomes, living deprivation, and urbanisation. Higher index represents more wellbeing and access to services [24] | Clinical Records |
| Family Ed | 1) Unknown, 2) Primary School Uncompleted, 3) Primary School Completed, 4) Secondary Uncompleted, 5) Secondary Completed, 6) Training Uncompleted, 7) Training Completed (In Chile from 2 to 3 years of formal education), 8) University Uncompleted, 9) University Completed (In Chile from 4 to 7 years of formal education) | DHH Survey |
| Add diff | 1) Not declared, 2) Declared. | Clinical Records |
| CI condition | 1) Operative-Without issues, 2) Operative but with some issues, 3) Not Operative - Technical Issues, 4) Not in use from time ago. | DHH Survey |
| CI status | 1) Unilateral CI w/o contra HA, 2) Unilateral CI with contra HA, 3) is this bilateral? | DHH Survey |

*(Continued)*

**Table 1.** (Continued)

| Variable | Description | Source |
|---|---|---|
| Rehabilitation attendance | 1) Attending, 2) Attending with difficulties, 3) No attending. | DHH Survey |
| Frequency rehabilitation | 1) Weekly, 2) Each 2 weeks, 3) Monthly, 4) Less than once per month. | DHH Survey |
| Duration Treatment | *Three levels from the question; How much time do you spend in each session? 1)* 1 hour, 2) 30 min, 3) Less than 30 min, 4) 40-45 min, 5) More than 2 hrs. | DHH Survey |
| Commute time | 1) Very short time, 2) Short Time, 3) Enough Time, 4) Long Time, 5) Very long Time, 6) Not attending | DHH Survey |
| Education attendance | 1) Not attending any ed. 2) Special school for the deaf, 3) Mainstream ed. w/o SNA, 4) Mainstream ed. with SNA, 5) Special school for SLD, 6) Mainstream nursery. | DHH Survey |
| CI use (ordinal scale) | 1) Never, 2) Sometimes, 3) Frequently, 4) Always | DHH Survey |
| CI hrs per day (minutes) | Minutes per day. | DHH Survey |
| Behavioural problems | Two level: 1) No, 2) Yes | DHH Survey |
| Treatment Professionals | Multiple selection variable from the question; *If he/she receives support for special needs with CI, who delivers this support?* 1) Interpreter, 2) Speech and Language Therapist, 3) ENT/Doctor & Audiologist, 4) Special Educator, 5) AVT Therapist, 6) Psychologist, 7) Unknown, Not remember, 8) Other. | DHH Survey |
| CI Training | 1) No, 2) Yes | DHH Survey |
| CI Confidence | 1) No confidence, 2) Poor Confident, 3) Enough Confident, 4) Very Confident, 5) Very Poor Confident | DHH Survey |

Notes: Abbreviations: DHH Survey = Online survey to participants, Hospital C. Records = Hospital Clinical Records, GeersM = Geer and Moog Latin American Categories of Speech Perception, CAPII = Chilean Categories of Auditory Performance CAP II, SIR = Chilean Speech Intelligibility Rating Scale SIR (Chilean Version of SIR), SHI = Socioeconomic Health Insurance level, BDI = Borough Development Index, Ed = Education, Add Diff = Additional Difficulties, Late Deafness ^ = Deafness occurring after 2 year of age. CI = Cochlear Implant, *CIuse = variable of CI use hours during the day, HA = Hearing aid/s, SNA = Special Needs Assistance, ENT = Ear, Nose and Throat, AVT = Auditory Verbal Therapy, CI Training: Parental Engagement variable about previous parental training with the CI, CI Confidence: Parental Engagement variable about confidence with the CI intervention.

## Outcomes

**National characterisation analysis.** We categorised participants based on their residence location and sociodemographic characteristics. Initially, their living areas were plotted on a map of Chile, with each national borough shaded according to its BDI (refer to Fig 1). Geographical distribution of deaf children with CI (N = 107) by BDI in Chile. Later, a descriptive analysis of independent variables of the sample was completed: continuous variables were summarised using means (medians), standard deviation (SD), and interquartile range (IQR), while categorical and binary variables were described by their frequencies and percentages within each category.

**Impact of variables in the outcomes in children with CI.** To evaluate the impact of potential sociodemographic and treatment variables on the outcomes expected with CI, we examined the association between nine selected variables: Children Age, CI Age, Additional Difficulties, Socioeconomic Health Insurance (SHI), Borough Development Index (BDI), Family Education, CI use, CI Training, CI Confidence– and our six outcomes results: 1) Communication at home, 2) Geer and Moog Latin American Categories of Speech Perception (GeersM), 3) Chilean Version of the Categories of Auditory Performance (CAPII), 4) Chilean Version of the Speech Intelligibility Rating Scale (SIR), 5) CI Satisfaction 6) Social Inclusion). We used non-parametric tests such as Spearman correlations and Wilcoxon t-tests. To account for missing data, we employed multivariate normal distribution methods to minimise biases

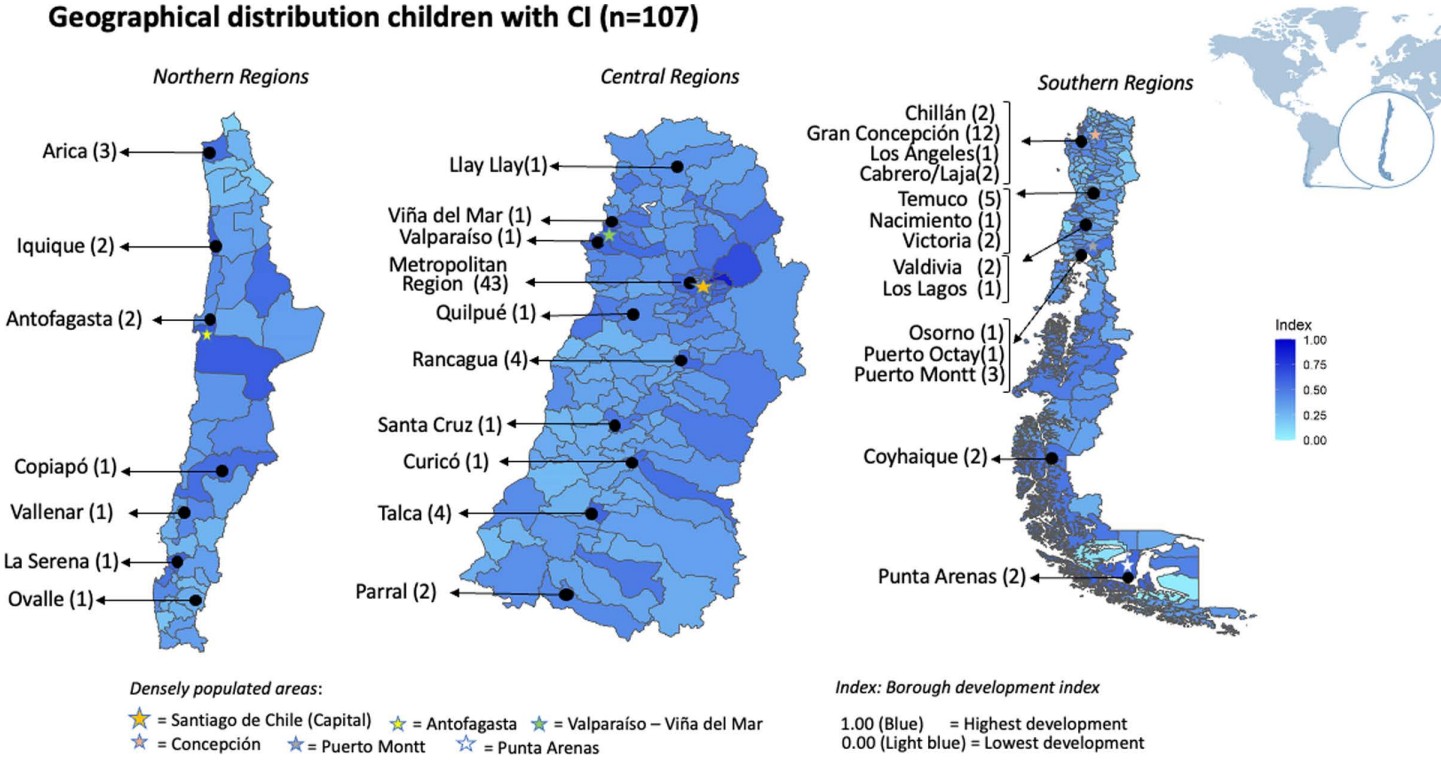

**Fig 1. Geographical distribution of deaf children with CI (N=107) by BDI in Chile.**

while obtaining appropriate estimates for uncertainty using the "mice" package in R Studio. This method was applied to two of the predictor factors - BDI and SES -, which had less than 3% of missing data, and GeersM results, which had 13% of random missing data. Additionally, to avoid estimation biases, we examined factor distribution and tested for multicollinearity, using the Variance Inflator Factor (VIF). We examined collinearity among our included variables; variables with a variance inflator factor (VIF) > 5 were removed. All statistical analyses were performed using R Studio version 1.4.

## 3. Results

### Descriptive analysis

Fig 1 illustrates the geographical distribution of our sample by Chilean boroughs using the BDI scores (from 0.0 to 1.0). Most of the children with CI came from the central area (55/107, 51.4%), specifically Santiago (43/107, 40.1%), and southern regions (37/107, 35.5%). Santiago and Concepcion have the highest BDI scores (0.78 and 0.64, respectively), whereas the remaining areas ranged below 0.60. The average BDI value in our sample was somewhat higher than the country average (0.54 vs. 0.37, respectively).

Table 2 shows the descriptive statistics for all sociodemographic, audiological and treatment variables for the sample, ranging in age from 2 years 4 months to 15 years 1 month, with an average age of 7 years. The sample was slightly more female (56%). Most children (81%) had congenital deafness, due to factors such as idiopathic causes, prematurity, or genetic conditions. Notably, 89.7% of these children did not face additional difficulties in addition to their deafness.

**Table 2. Independent variables descriptive statistics in deaf children with CI (N = 107).**

| Independent variable | Category/Level | Mean/Freq. [SD](%) |
|---|---|---|
| Age | Years and months of age. From min = 2y 4m, max = 15y 1m | 7 years, 0 months [2.9] |
| Gender | Male | 47 (44%) |
| CI age | Months of age at CI surgery | 33.19 [10.2] |
| Deafness Presentation | Congenital | 89 (81%) |
| | Late Deafness | 18 (19%) |
| SHI | Low income | 42 (39.3%) |
| | Low-middle income | 28 (26.2%) |
| | Middle income | 16 (15%) |
| | Middle - high Income | 21 (19.6%) |
| BDI | Index from 0 up to 1 | 0.54 [0.1] |
| Family Ed | Primary incompleted | 4 (3.7%) |
| | Primary completed | 7 (6.5%) |
| | Secondary incompleted | 46 (43%) |
| | Secondary completed | 9 (8.4%) |
| | Training incompleted | 12 (11.2%) |
| | Training completed | 5 (4.7) |
| | University incompleted | 16 (15%) |
| | University completed | 8 (7.5%) |
| Add diff | Not recorded/ Not declared | 96 (89.7%) |
| CI status | Unilateral CI w/o contra HA | 83 (77.6%) |
| | Unilateral CI with contra HA | 10 (9.3%) |
| | Bilateral CI | 6 (5.6%) |
| | Other type | 8 (7.5%) |
| CI condition | Operative-Without issues | 71 (66.4%) |
| | Operative but with some issues | 29 (27.1%) |
| | Not Operative – Technical issues | 3 (2.8%) |
| | Not in use from time ago | 4 (3.7%) |
| Rehabilitation attendance | Attending | 69 (64.5%) |
| | Attendance with difficulties | 25 (23.3%) |
| | Not attending | 13 (12.1%) |
| Easy rehabilitation attendance | Yes | 76 (71%) |
| | No | 31 (29%) |
| Frequency rehabilitation | Weekly | 78 (72.8%) |
| | Each 2 weeks | 10 (9.3%) |
| | Monthly | 7(6.5%) |
| | Less than once per month | 12 (11.2%) |
| Duration treatment | More than 2 hours | 2 (2.8%) |
| | 1 hour | 31 (29%) |
| | 40-45 minutes | 46 (43%) |
| | 30 minutes | 26 (24.3%) |
| | Less than 30 minutes | 1 (0.9%) |
| Education attendance | Not attending any ed. | 10 (%) |
| | Special school for the deaf | 12 (12.2%) |
| | Mainstream ed. w/o SNA | 6 (4.1%) |
| | Mainstream ed. with SNA | 53 (59.2%) |

*(Continued)*

**Table 2.** (Continued)

| | | |
|---|---|---|
| | Special school for SLD | 6 (10.2%) |
| | Mainstream nursery | 13 (2.0%) |
| CI use (ordinal variable) | Never | 2 (4.1%) |
| | Sometimes | 1 (2.0%) |
| | Frequently | 5 (10.2%) |
| | Always | 41 (83.7%) |
| CI hrs per day | From min = 0.0 max = 19.0 hrs. | 10.5 [3.6] |
| Behavioural challenges | No | 86 (80.4%) |
| | Yes | 21 (19.6%) |
| Treatment Professionals | Interpreter | 9 (8.4) |
| | Speech & Language Therapist | 89 (83.1) |
| | ENT/Doctor or Audiologist | 50 (46.7) |
| | Special Educator | 54 (50.5) |
| | AVT Therapist | 28 (26.1) |
| | Psychologist | 26 (24.2) |
| | Other | 8 (7.5) |
| Parental Engagement: | No | 34 (31.8%) |
| • CI Training | Yes | 73 (68.2%) |
| Parental Engagement: | No confidence | 1 (0.9%) |
| • CI Confidence | Very Poor Confident | 4 (3.7%) |
| | Poor Confident | 7 (6.5%) |
| | Somewhat Confident | 41 (38.3%) |
| | Very Confident | 54 (50.5%) |

Notes: Abbreviations: NR = Not recorded, SHI = Socioeconomic Health Insurance level, BDI = Borough Development Index, Ed = Education, Add Diff = Additional Difficulties, Late Deafness = Deaness with onset after birth, CI = Cochlear Implant, DHH Survey = Child with CI Survey 1,Tech. = Technical, *CIuse = variable of CI use hours during the day, HA = Hearing aid/s, SNA = Special Needs Assistance, SLD = Speech and Language Disorders, Previous training variable, ENT = Ear, Nose and Throat, AVT = Auditory Verbal Therapy, CI Training: Parental Engagement variable about previous parental training with the CI, CI Confidence: Parental Engagement variable about confidence with the CI intervention.

Socioeconomic analysis revealed that 42% of the children came from low-income families, and overall, 65% were from the two lowest-income levels in the public health system. Parental education was relatively high, with 81% of families having at least one parent who completed secondary education and 22% having a parent with a higher level of education.

The age at which children underwent CI surgery varied from 1 year to 12 years 4 months, with an average age of 4 years 6 months. For children with congenital deafness, the mean age at surgery was younger, at 3 years 10 months. The majority (77.6%) of children used a unilateral CI without a hearing aid in the other ear. Nearly all children (94%) were receiving treatment from speech-language therapists or other professionals, primarily on a weekly basis (78%). Most children (83.1%) attended speech and language therapy, and only a small fraction (8.4%) used a sign language interpreter during intervention.

Regarding educational engagement, 90.7% of children were attending school or nursery, and a significant majority (78.5%) of parents reported that their children used their CI consistently, with an average daily use of 10.53 hours. Behavioural issues were uncommon, with 80% of parents reporting no problems. Parental engagement was substantial, with 94% having

received specific training for CI management and 89% expressing high to very high confidence in handling the device

Table 3 describes the outcomes of deaf children with CI in Chile (N = 107), in terms of their communication, auditory performance, and speech intelligibility, parental satisfaction and social inclusion.

Regarding communication at home, 56% of the children used a combination of signs and spoken language, while 32% communicated exclusively through spoken language. This mixed approach indicates a reliance on both visual and auditory methods to facilitate communication within families. The Geers and Moog Categories assessment revealed that 55% were in the lowest two categories, indicating limited ability to perceive speech even with amplification. Only 12.1% achieved significant speech comprehension and word recognition scores.

CAPII results showed that 62.6% were rated in the higher categories [5,7,8], indicating a good understanding of common phrases without needing to lip-read. On the other hand, in SIR results, 48.6% of the parents rated their child's speech as either "intelligible" or "completely intelligible." However, 38.3% indicated their child's connected speech was unintelligible and often relied on manual communication.

Moreover, 70.1%, of parents were "Satisfied" or "Very Satisfied" with the CI's impact on their child's life, although 5.6% reported being "Very Unsatisfied." Despite differing abilities in speech perception and spoken language, 70.1% of parents rated their child's inclusion in school and social life at the highest levels.

## Statistical analysis

Table 4 displays the results of the inferential analyses for the associations between the outcome variables and sociodemographic and treatment variables in deaf children with CI (n = 107). We found positive associations between the children's age and three of the outcomes: Communication at Home (p = 0.006*), GeersM (p = 0.003*) and SIR (p = 0.001*). There were also positive correlations between the CI Age and two outcomes, GeersM (p = 0.006*) and CAPII (p = 0.058). The Borough Development Index was associated with the Communication at Home (p = 0.008*). In addition, CI use had positive correlations with CAPII (p < .001*) and SIR (0.001*).Finally, CI training as a binary variable had a positive effect on the GeersM scale (p < .001*).

In contrast, no significant correlations were found between the Social Health Insurance (SHI) of deaf children with CI and any of the measured outcomes. Similarly, Family Education was not significantly associated with any of the outcome results. Finally, CI Confidence did not have any associations with the measured outcomes.

## 4. Discussion

This study is the first to profile deaf children with CI in Chile and the Latin American region, covering 70% of those implanted from 2017 to 2019. For socioeconomic characteristics, our findings revealed the diversity of deaf children receiving CI, the majority of whom belong to middle- and low-income families. Additionally, participants with training or university completed (26%) were lower than the average of high-income countries (39%). This may reduce the likelihood of children having access to a rich communicative environment and favourable conditions for receiving a CI, which, in turn, might impact their development of spoken language skills.

The average BDI in the characterisation study was .54, which was higher than the national average of .37. Thus, the children in this study had more advantageous conditions than the average population in Chile. Consistent with Madriz [8], findings suggest that children with

**Table 3. Descriptive statistics outcomes in deaf children with CI (N = 107).**

| Dependent variable | Median/Category | Mean/Freq. | % |
|---|---|---|---|
| **Communication at home** | Mixed Language | 60 | 56 |
| | Spoken Language | 34 | 31.7 |
| | Other | 7 | 6.5 |
| | Sign Language | 6 | 5.6 |
| **GeersM** | 0.Unable to perceive any speech sound (ling test) | 7 | 6.5 |
| | 1.Speech detection but unable to perceive even time-intensity pattern info in words with speech amplified | 34 | 31.8 |
| | 2.Time-intensity pattern perception in amplified speech (Above 70% in testing) | 25 | 23.4 |
| | 3.Word identification by basic spectral information (Same metric and time, but different consonants and vowels) | 10 | 9.3 |
| | 4.Word identification in a group by their vowels | 5 | 4.7 |
| | 5.Recognition of word in a group by their consonants | 13 | 12.1 |
| | 6.Word recognition by repetition | 7 | 6.5 |
| | 7.Spoken language comprehension | 6 | 5.6 |
| **CAPII** | 0. No awareness of environmental sounds | 4 | 3.7 |
| | 1. Awareness of environmental sounds | 2 | 1.9 |
| | 2. Response to speech sounds | 6 | 5.6 |
| | 3. Recognition of environmental sounds | 8 | 7.5 |
| | 4. Discrimination of at least two speech sounds | 12 | 11.2 |
| | 5. Understanding of common phrases without lipreading | 21 | 19.6 |
| | 6. Understanding of conversation without lipreading with a familiar talker | 5 | 4.7 |
| | 7. Use of a telephone with a familiar talker | 19 | 17.8 |
| | 8. Understanding/Following group conversations. | 23 | 21.5 |
| | 9. Use the telephone with an unknown speaker in an unpredictable context. | 7 | 6.5 |
| **SIR** | 1. Connected speech is unintelligible. Pre-recognizable words in spoken language, the child's primary mode of everyday communication may be manual. | 41 | 38.3 |
| | 2.Connected speech is unintelligible; intelligible speech is developing in single words when context and lip-reading cues are available | 7 | 6.5 |
| | 3.Connected speech is intelligible to a listener who concentrates and lip-reads within a known context. | 7 | 6.5 |
| | 4.Connected speech is intelligible to a listener who has little experience of a deaf person's speech. | 30 | 28 |
| | 5. Connected speech is intelligible to all listeners. The child is understood easily in everyday contexts. | 22 | 20.6 |
| **CI Satisfaction** | No, not too much included | 6 | 5.6 |
| | No, not included | 5 | 4.7 |
| | Neutral | 21 | 19.6 |
| | Yes, the child is included | 24 | 22.4 |
| | Yes, the child is quite included | 51 | 47.7 |
| **Social Inclusion** | No, not too much included | 7 | 6.5 |
| | No, not included | 4 | 3.7 |
| | Neutral | 15 | 14 |
| | Yes, the child is included | 22 | 20.6 |
| | Yes, the child is quite included | 59 | 55.1 |

Notes: In Communication at Homme "Mixed Language" includes spoken and sign language, "Other" includes any other predominant communication at home. Abbreviations: GeersM = Geer and Moog Latin American Categories of Speech Perception, CAP II = Categories of Auditory Performance, SIR = Speech Intelligibility Rating Scale Categories.

**Table 4. Inferential analyses for the association between our outcomes and sociodemographic and treatment variables in deaf children with CI (n = 107).**

| Predictors | Outcomes | | | | | |
| --- | --- | --- | --- | --- | --- | --- |
| | Communication at h. | GeersM | CAPII | SIR | CI Satisfaction | Social Inclusion |
| Children Age | **X² = 836, p = 0.006*** | **cor: 0.277, p = 0.003*, CI = 0.147** | **cor: 0.251, p = 0.009, CI = 0.083** | **cor: 0.311, p = 0.001*, CI = 0.163** | cor: 0.015, p = 0.87, CI = -0.0162 | cor: 0.039, p = 0.693, CI = -0.213 |
| CI Age | X² = 734, p = 0.3447 | **cor: 0.260, p = 0.006*, CI = 0.136** | **cor: 0.184, p = 0.058*, CI = 0.036** | cor: 0.252, p = 0.009, CI = 0.133 | cor: 0.021, p = 0.830, CI = -0.146 | cor: 0.012, p = 0.902, CI = -0.223 |
| Additional Difficulties | X² = 3.71, p = 0.7245 | **X² = 127.31, p<.001*** | X² = 384, p = 0.8903 | X² = 518.5, p = 0.1309 | X² = 412. p = 0.8442 | X² = 502.5, p = 0.658 |
| SHI | X² = 991, p = 0.080 | cor: 0.021, p = 0.823, CI = -0.208 | cor: 0.022, p = 0.825, CI = -0.200 | cor: -0.042, p = 0.664, CI = -0.223 | cor: -0.103, p = -0.289, CI = -0.298 | cor: -0.015, p = 0.876, CI = -0.207 |
| BDI | **X² = 849, p = 0.008*** | cor: 0.095, p = 0.350, CI = 0.067 | cor: 0.096, p = 0.324, CI = -0.105 | cor: 0.159, p = 0.102, CI = 0.039 | cor: 0.040, p = 0.684, CI = -0.240 | cor: 0.045, p = 0.647, CI = -0.144 |
| Family Education | X² = 1245, p = 0.9777 | cor: 0.043, p = 0.655, CI = -0.161 | cor: 0.007, p = 0.704, CI = -0.133 | cor: 0.159, p = 0.974, CI = -0.199 | cor: 0.028, p = 0.774, CI = -0.176 | cor: 0.045, p = 0.103, CI = -0.026 |
| CI use | X² = 1002, p = 0.104 | cor: 0.125, p = 0.200, CI = -0.063 | **cor: 0.339, p<.001*, CI = 0.205** | **cor: 0.314, p = 0.001*, CI = 0.034** | cor: 0.251, p = 0.009, CI = 0.101 | cor: 0.185, p = 0.057, CI = -0.173 |
| CI Training | X² = 1093, p = 0.2325 | **X² = 83.593, p<.001*** | X² = 117, p = 0.425 | X² = 118, p = 0.451 | X² = 105, p = 0.098 | X² = 119, p = 0.478 |
| CI Confidence | X² = 935, p = 0.02569 | cor = 0.05, p = 0.573, CI = -0.103 | cor = 0.300, p = 0.427, CI = 0.144 | cor = 0.073, p = 0.472, CI = -0.078 | cor = 0.161, p = 0.098, CI = 0.129 | cor = 0.116, p = 0.479, CI = 0.049 |

Notes: Abb: Communication at h. = Communication at home. Abbreviations: X²: Wilcoxon Test, cor=correlation, CI = Confidence Interval, CI age = Age at CI implantation, SHI = Socioeconomic Health Insurance level, BDI = Borough Development Index, Family Ed = Highest education parent/caregiver, CIuse = variable of CI use hours during the day, CI = Cochlear Implant, GeersM = Geer and Moog Latin American Categories of Speech Perception, CAPII = Categories of Auditory Performance, SIR = Speech Intelligibility Rating Scale Categories. **Alpha** = p-value **0.05 '*'**

CI and other assistive technology users, on average, originate from more developed boroughs compared to the national average. This inequality, which affects CI candidates from less developed boroughs, emphasises the need to focus on deaf children from less developed boroughs, ensuring equal access to CI thus improving Universal Health Coverage. Thus, social determinants of health, including healthcare access/quality, education access/quality, social and community context, economic stability, and neighbourhood environment, can directly influence the expected reach and results of the CI policy covering deaf children at an early age.

In the audiological characterisation, the vast majority of children with CI were reported to have congenital rather than late onset deafness. Additionally, a low percentage (8%) of children were reported to have additional difficulties. This percentage is lower than the 30% of children reported with additional difficulties in previous studies worldwide [25]. Although the low representation in Chile may be due to the limited inclusion of children with additional difficulties in the CI policy, this could also be partially explained by the under diagnosis of some additional developmental disabilities in deaf children [25].

It is important to note the low percentage of children with CI using a contralateral hearing aid with unilateral CI (10%) or bilateral CI (6%). Although only unilateral CI has been evaluated as cost-effective in low and middle income countries in the Latin American region, it is likely that using contralateral hearing aids with a unilateral CI, or using bilateral CI would positively affect spoken language outcomes.

In Chile, the total cost of CI rehabilitation (US$52250) (Appendix "S4") is in line with other high-income countries, where total costs stand at US$61000 in Singapore, US$55000 in the UK and US$52,000 in Switzerland [26]. While the CI has been deemed cost-effective in certain middle-income nations [12], a comprehensive assessment of its policy benefits in Chile is necessary. Factors such as inequalities, insufficient funding, and limited social protection programs could impact its cost-effectiveness. Considering the apparent lack of CI coverage

for those deaf children from less developed boroughs, it is crucial to include them to properly evaluate whether this high-cost intervention in Chile is cost-effective.

In terms of treatment characteristics, parents/caregivers reported high attendance (94%) regarding CI treatment and education. This is particularly positive for the Chilean public health system meaning that most deaf children with CI received treatment despite difficulties attending the centres, especially during the COVID-19 pandemic.

The study found that children used their CI for an average of 10.5 hours per day, indicating high usage, while parental reports in previous studies have varied. Reported averages range from 7.6 to 12 hours, highlighting the need for objective measures like device data logging [27]. In parental engagement, most parents/caregivers received some training about the device, and also showed high CI confidence. This is a positive observation, particularly since rehabilitation practices typically prioritise the child's needs, often overlooking the needs of parental support. Further studies might need to assess the parental training and health education in the CI results.

Only a third of parents/caregivers reported that their deaf child used spoken language at home and with others. Thus, it is important to consider this low percentage. Results from the CAP II showed that more than half of the sample could understand common phrases and conversations through spoken language, while from the SIR, only around half of parents/caregivers reported that their children's speech was intelligible to a listener with experience with deaf people. Additionally, more than half of the participants were categorised in the lowest ranking in the GeersM results. These disparities in speech perception and language outcomes can be explained by complications in measuring language development in deaf children, which depends on factors such as socioeconomic status, language proficiency, dialect, and the child's level of deafness, combined with the fact that some of the language assessments were conducted by professionals and others by parents/caregivers, [28]. In addition, although these assessments evaluate key aspects of speech perception and production, they also measure varying aspects of language ability, and this variation might also contribute to the discrepancies between parent reported and clinically-reported outcomes.,.

The GeersM results from the clinical records showed than almost 50% of deaf children with CI completed only the lowest level for speech perception. Although the evaluation was taken after 12 months of CI use, this percentage is low compared with previous studies [22]. This finding reinforces the need for measures in low- and middle-income settings that face similar development challenges, especially since over 80% of children with disabilities live in those regions.

In satisfaction outcomes, most parents were very satisfied with their child's CI, which is interesting considering the variation in the spoken language outcomes with the CI. A possible explanation for this is that a high level of satisfaction by parents/caregivers could be due to courtesy bias or a lack of criticism because the services are mainly provided free of charge in the public system [16]. This is a risk when the success of the CI is measured only in terms of satisfaction. Although satisfaction is important, the focus should remain on the main objective of providing CI for deaf children, which is spoken language development.

The inferential analysis shed light on potential factors that affect outcomes for deaf children with CI in Chile. As expected, older children with CI achieved higher scores on some of the outcomes because of the wide age range of participating children from two to fifteen years old. Additionally, approximately 19% of children in the sample had late deafness, with likely some spoken language development preceding the onset of deafness. In addition, and along with previous evidence, children with additional difficulties achieved lower rankings in the scale results compared to those without additional difficulties [18].

As in previous studies, higher CI use in deaf children emerged as a critical measure related to better performance on CAPII and SIR outcomes [28]. Similarly, the finding that children of parents/caregivers who received training about CI had higher GeersM scores than those who

reported no training are in line with previous evidence of a positive effect on spoken language outcomes [22]. Thus, CI use and CI training might be translated to concrete measures to be recorded during the rehabilitation process.

Borough Development Index also had a negative influence as a social determinant of health on the outcome measure of Communication at Home. Restricted essential services in healthcare, education, and rehabilitation without adequate infrastructure and proximity are still significant obstacles faced by children with CI in less developed boroughs in Chile. Previous studies have suggested that multidimensional aspects of social and economic conditions in a family can deter the future with CI, especially in unequal contexts [29].

In this study, satisfaction and social inclusion were not significantly influenced by any of the potential factors. In addition, the child's age can be detrimental in the parental evaluation of these outcomes results. Further research could evaluate satisfaction in deaf children after longer term use of the device, and social inclusion for deaf children with CI who have been attending mainstream nurseries/schools for deaf children.

This study has some limitations. First, although data for one of the main outcomes was obtained from clinical records, the majority of data were provided by parents/caregivers, which can introduce potential bias in the study. Second, while we had a representative number of participants, the sample size and the number of factors under consideration could potentially affect our statistical analyses. Third, although this study represented 70% of those children implanted within the time period, further studies needs to consider specific calculations for the sample size, which are needed for a deeper analysis of the significance of the inferential results from the study. Nevertheless, this study does present some considerable strengths. It is the first evaluation of a national sample of deaf children with CI in the Latin American region, and this is the first report across a range of outcomes for children with CI in Chile, which will support the national effort for providing better services to deaf children. These results also have implications for countries in similar conditions providing CI.

## 5. Conclusion

This first characterisation of deaf children with CI in Chile offers comprehensive results considering demographic, socioeconomic, treatment variables, and six different expected outcome results with the CI. Despite presenting lower levels of performance relative to international benchmarks, spoken language outcomes do align with the regional context. The finding of relatively low performance among deaf children with CI pinpoints to an area of potential intervention and development, which needs to be assessed in detail in further studies. Several factors were identified as influencing the outcomes in deaf children with CI, including the child's age, a socio-economic factor, CI use and CI training. In addition, additional difficulties were found to be an influential negative variable in speech perception performance. Furthermore, borough development index as a social determinant of health was noted as having a positive influence on the speech perception in children with CI. Importantly, better use of the CI device and increased parental engagement also emerged as pivotal positive factors, which can be beneficial in implementing concrete targets in the rehabilitation process of deaf children with CI in Chile and potentially throughout Latin America.

## Supporting information

**S1 File. Ethic approval University of Chile (English translation).**
(PDF)

**S2 File. Content permission MAP.**
(DOCX)

**S3 File. Inclusivity in global research.**
(DOCX)

**S4 File. Appendix 1. Consort diagram for participant selection. Appendix 2.** Informed consent. **Appendix 3.** Protocols. **Appendix 4.** Analysis of total cost of CI in Chile.
(DOCX)

**S5 File. Figure 1 with PACE adjustment.**
(DOCX)

## Acknowledgments

All authors attest they meet the ICMJE criteria for authorship and have reviewed and approved the final article. As a result of restrictions from March 2020, all measures were taken to follow the national guidance on research in health services.

## Author contributions

**Conceptualization:** Mario Bustos-Rubilar, Fiona Kyle, Merle Mahon.

**Data curation:** Mario Bustos-Rubilar.

**Formal analysis:** Mario Bustos-Rubilar, Merle Mahon.

**Funding acquisition:** Mario Bustos-Rubilar, Merle Mahon.

**Investigation:** Mario Bustos-Rubilar, Fiona Kyle, Merle Mahon.

**Methodology:** Mario Bustos-Rubilar, Fiona Kyle, Merle Mahon.

**Project administration:** Mario Bustos-Rubilar, Fiona Kyle, Merle Mahon.

**Resources:** Mario Bustos-Rubilar.

**Software:** Mario Bustos-Rubilar.

**Supervision:** Mario Bustos-Rubilar, Fiona Kyle, Merle Mahon.

**Validation:** Mario Bustos-Rubilar, Fiona Kyle.

**Visualization:** Mario Bustos-Rubilar.

**Writing – original draft:** Mario Bustos-Rubilar.

**Writing – review & editing:** Mario Bustos-Rubilar, Fiona Kyle, Merle Mahon.

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
