## [Decision Letter · Decision Letter 0]

12 Nov 2024

Dear Dr. Bustos-Rubilar,

Thank you for submitting your manuscript to PLOS ONE. After careful consideration, we feel that it has merit but does not fully meet PLOS ONE’s publication criteria as it currently stands. Therefore, we invite you to submit a revised version of the manuscript that addresses the points raised during the review process.

Take into account reviewers concerns

Rewrite the discussion section in order to focus more precisely in the results of the manuscript

We look forward to receiving your revised manuscript.

Kind regards,

Paul H Delano, Ph.D.

Academic Editor

PLOS ONE

“the data collection was partially supported by a small grant from the Chilean Ministry of Health through the National Public Tender 757-89-l120.”

“All authors attest they meet the ICMJE criteria for authorship and have reviewed and approved the final article. This article was supported by a full scholarship provided by the Chilean Government "Beca de Doctorado en el Extranjero Becas Chile, Convocatoria 2018, Ley N°21.053, Asociación Nacional de Investigación y Desarrollo (ANID)" and the Chilean Ministry of Health through the National Public Tender 757-89-l120.”

“the data collection was partially supported by a small grant from the Chilean Ministry of Health through the National Public Tender 757-89-l120.”

5. In the online submission form, you indicated that [Data are available upon appropriate request].

7. We note that Figure 1 in your submission contain [map/satellite] images which may be copyrighted. All PLOS content is published under the Creative Commons Attribution License (CC BY 4.0), which means that the manuscript, images, and Supporting Information files will be freely available online, and any third party is permitted to access, download, copy, distribute, and use these materials in any way, even commercially, with proper attribution. For these reasons, we cannot publish previously copyrighted maps or satellite images created using proprietary data, such as Google software (Google Maps, Street View, and Earth). For more information, see our copyright guidelines: http://journals.plos.org/plosone/s/licenses-and-copyright.

Additional Editor Comments:

Take into account reviewers concerns

rewrite the discussion section in order to focus more precisely in the results of the manuscript

Reviewers' comments:

Reviewer's Responses to Questions

**Comments to the Author**

1. Is the manuscript technically sound, and do the data support the conclusions?

Reviewer #1: Yes

Reviewer #2: Yes

2. Has the statistical analysis been performed appropriately and rigorously?

Reviewer #1: Yes

Reviewer #2: Yes

3. Have the authors made all data underlying the findings in their manuscript fully available?

Reviewer #1: No

Reviewer #2: Yes

4. Is the manuscript presented in an intelligible fashion and written in standard English?

Reviewer #1: Yes

Reviewer #2: No

Reviewer #1: The study is interesting, important, relevant, and novel. It is well written, the reading is easy.

The main problem of the study is that: In the discussion there is an exploratory analysis and review of the cost of cochlear implants what is not directly related to the aim of the study. In this study the authors could use it as an example or for argumentation just to cite it the total cost cost but not to analyze extensively as the authors did in the discussion of the paper, since it is not the objective of the study. I would remove the extensive discussion and analysis of costs included in the discussion. Remove it from the discussion or just add it to supplementary, exploratory analysis. It is unnecessary, it is not mentioned in the abstract, not related to the main question of the study and also not related to the message that apparently the authors want to give. The authors state: “This study will characterize deaf children with CI in Chile and evaluate the impact of the CI in speech perception and production, social inclusion, and parental satisfaction.” The study is not about the total cost of CI

The discussion should present and discuss the main result, discuss other results related to the objectives, but not results not related to the objectives. The authors mainly discuss and repeat their results, and in the end they spoke about few limitations, actually recall bias is recognized but not named. They do not go deeply into the sample size although they recognize it might be not sufficient or be the reason for lack of significance. The sample size was not calculated, although the sample was about the 70 % of total CI. The future studies are not mentioned at the end of discussion but some were mentioned when discussing some results

The conclusion is correct but I would emphasize more about the take home message by ending the conclusion paragraph on that phrase, which is nearly but not the last phrase of the paper.

In variables characterization aethiology does not sound fine for congenital or late hearing loss, that is moment of presentation, I think it is more correct instead of aethiology. (see Table 1)

The abstract is good.

The methods are fine. A missing data of 3% reported is acceptable and they report methods of missing data management.

Reviewer #2: The article “Deaf children with cochlear implants in Chile: A national analysis of health determinants and outcomes in the Latin American context” is interesting and worth publishing, but needs some revisions.

Authors provide a description of a group of pediatric patients receiving cochlear implants in Chile, the first report of the country. In order for readers to understand some results I would suggest the authors to include a description of the cochlear program and how patients are selected. For example, if the patient has a deprived social context without family support he/she might be rejected from candidacy. This observation has a direct impact on the social description of the sample. Another issue is the observation that the majority of children with cochlear implants live either in Santiago, capital city, or Concepcion, second largest city of the country. Public hospitals authorized by the Health Minister to perform cochlear implant surgery are also located in these two cities, so maybe ENT from these geographical locations are more prone to propose candidates than ENT in smaller cities without cochlear implant programs.

Other observations that should be addressed by the authors:

Detection of hearing loss in newborns is mandatory since January 2020 for babies delivered in public hospital.

Authors excluded cochlear implants cases from private institutions, why?

In methods please explain socioeconomic health insurance. In Chile health provision by the government is different according to income, and people pay a percentage according to a classification (ranging from 0 to 50%). This classification might not reflect the total home income since it does not consider informal income.

Speech perception according to clinical records (Peer and Moog) differed from parent's perception (CAPII). For the clinical assessment 55% were in the two lowest categories, on the contrary, parents perceive that 62.6% were in the higher categories. Authors should comment on this discrepancy.

Include intervals of confidence for the correlation factors.

There is a table in the discussion that is out of place. The content was not included in the objectives nor in the results.

Revise grammar, for example page 10 line 259, page 11 second paragraph, page 14 line 274.

**Do you want your identity to be public for this peer review?** For information about this choice, including consent withdrawal, please see our Privacy Policy

Reviewer #1: No

Reviewer #2: **Yes: ** Mariela C. Torrente

---

## [Author Response · Author response to Decision Letter 1]

17 Dec 2024

Dear Editor,

Thank you very much for your email. We have removed all document duplicated or old versions from the submission. Thank you very much for your advice.

Regarding your questions about the image,

-Where did the authors obtain the map, basemap, shapefile, map data, etc in Figure 1?

The map was created by the research team using RStudio with the CRAN package and later modified using Microsoft Office.

-What software was used to make this map?

R Studio with CRAN package:

https://cran.r-project.org/web/packages/chilemapas/index.html

Microsoft Office with Power Point:

https://www.microsoft.com/es-cl/microsoft-365/powerpoint

Just in case, we have attached the content permission form.

Thank you very much.

Best wishes,

Mario

TO reviewers:

Reviewer #1 Comments:

10. General Feedback

Thank you very much for your encouraging words.

11. Discussion on Cost Analysis

We agree that the cost analysis section was excessive. We have moved this discussion to the supplemental materials (Appendix 4), mentioning it only briefly as a reference in the main text.

12. Sample Size Discussion

We have included a comment on sample size limitations in the last paragraph of the discussion (lines 454-457).

13. Conclusion Emphasis

We have revised the conclusion to emphasize the main take-home message (lines 474-481).

14. Use of ‘Aetiology’

We have changed "aetiology" to "HL Presentation" in Tables 2 and 3, as suggested.

Reviewer #2 Comments:

15. General Feedback

Thank you very much for your consideration.

16. Cochlear Implant Program Description

We have added a description of the cochlear implant program and selection criteria in paragraph 5 of the introduction (lines 51-54).

17. Geographic Distribution of Patients

We have clarified the information regarding the locations of public hospitals authorized for cochlear implant surgery in paragraph 6 (lines 56-59).

18. Newborn Hearing Screening

We added this information to the introduction (line 59) and the Methods section, under Population (line 108).

19. Exclusion of Private Institution Cases

We clarified the exclusion criteria in the Methods section, under Population (lines 111-114).

20. Socioeconomic Health Insurance

We added a description of socioeconomic health insurance classification in Table 1.

21. Discrepancies in Speech Perception Ratings

We have discussed the discrepancy in parental vs. clinical ratings in the discussion section, paragraph 7 (lines 397-405).

22. Confidence Intervals

We have included confidence intervals in Table 4.

23. Out-of-Place Table in Discussion

We have addressed this in our response to point 11 above.

24. Grammar Revisions

We have addressed the suggested grammar improvements on pages 10, 11, and 14.

Finally, as a grammatical improvement, we have changed the term hearing loss to deafness to make it simpler and more understandable for any reader, while supporting the reduction of potential ableist terminology when referring to people with disabilities.

Once again, we greatly value all your suggestions, which have helped us improve our manuscript.

Thank you very much, and best wishes,

Mario Bustos-Rubilar et al.

---

## [Editor Report · Decision Letter 1]

26 Dec 2024

Deaf children with cochlear implants in Chile: A national analysis of health determinants and outcomes in the Latin American context.

PONE-D-24-43191R1

Dear Dr. Bustos-Rubilar,

We’re pleased to inform you that your manuscript has been judged scientifically suitable for publication and will be formally accepted for publication once it meets all outstanding technical requirements.

Kind regards,

Paul H Delano, Ph.D.

Academic Editor

PLOS ONE
---

## [Editor Report · Acceptance letter]

PONE-D-24-43191R1

PLOS ONE

Dear Dr. Bustos-Rubilar,

I'm pleased to inform you that your manuscript has been deemed suitable for publication in PLOS ONE. Congratulations! Your manuscript is now being handed over to our production team.

Kind regards,

on behalf of

Dr. Paul H Delano

Academic Editor

PLOS ONE